# Staff perceptions towards virtual reality-motivated treadmill exercise for care home residents: a qualitative feedback study with key stakeholders and follow-up interview with technology developer

Hannah Louise Bradwell  ,[1] Leonie Cooper,[1] Katie Jane Edwards  ,[1] Rory Baxter,[1] Simone A Tomaz,[2] John Ritchie,[3] Swen Gaudl,[4] Alejandro Veliz-Reyes,[5] Gemma C Ryde,[6] Tanja Križaj,[1] Alison Warren,[1] Arunangsu Chatterjee,[7] Katharine Willis,[5] Richard Haynes,[3] Catherine H Hennessy,[8] Anna C Whittaker  ,[2] Sheena Asthana,[1] Ray B Jones  ,[1] On behalf of the GOALD project

**Correspondence to**
Dr Hannah Louise Bradwell;
hannah.bradwell@plymouth.ac.uk

## ABSTRACT

**Objectives** Health and care resources are under increasing pressure, partly due to the ageing population. Physical activity supports healthy ageing, but motivating exercise is challenging. We aimed to explore staff perceptions towards a virtual reality (VR) omnidirectional treadmill (MOTUS), aimed at increasing physical activity for older adult care home residents.

**Design** Interactive workshops and qualitative evaluation.

**Settings** Eight interactive workshops were held at six care homes and two university sites across Cornwall, England, from September to November 2021.

**Participants** Forty-four staff participated, including care home, supported living, clinical care and compliance managers, carers, activity coordinators, occupational therapists and physiotherapists.

**Interventions** Participants tried the VR treadmill system, followed by focus groups exploring device design, potential usefulness or barriers for care home residents. Focus groups were audio-recorded, transcribed verbatim and thematically analysed. We subsequently conducted a follow-up interview with the technology developer (September 2022) to explore the feedback impact.

**Results** The analysis produced seven key themes: anticipated benefits, acceptability, concerns of use, concerns of negative effects, suitability/unsuitability, improvements and current design. Participants were generally positive towards VR to motivate care home residents' physical activity and noted several potential benefits (increased exercise, stimulation, social interaction and rehabilitation). Despite the reported potential, staff had safety concerns for frail older residents due to their standing position. Participants suggested design improvements to enhance safety, usability and accessibility. Feedback to the designers resulted in the development of a new seated VR treadmill to address concerns about falls while maintaining motivation to exercise. The follow-up developer interview identified significant value in academia–industry collaboration.

## STRENGTHS AND LIMITATIONS OF THIS STUDY

⇒ A relatively large sample of participants in qualitative research provides a breadth of perceptions and data saturation.
⇒ Rigorous qualitative analysis on a novel topic by a research team with diverse (yet related to the topic) disciplinary backgrounds (including engineering, health and design).
⇒ Explored the perceptions of key stakeholders (health and social care staff).
⇒ However, there are limited interaction durations between participants and the technology.
⇒ No perceptions were gathered from older adult stakeholders.

**Conclusion** The use of VR-motivated exercise holds the potential to increase exercise, encourage reminiscence and promote meaningful activity for care home residents. Staff concerns resulted in a redesigned seated treadmill for those too frail to use the standing version. This novel study demonstrates the importance of stakeholder feedback in product design.

## BACKGROUND

### Introduction

Pressure on health and social care resources is increasing because of increased life expectancy and decreasing numbers of care workers.[1 2] Therefore, there has been a push for research and innovation into technologies to support healthy ageing.[3 4] Reducing loneliness and remaining active are key strands of improved well-being for older adults.[5 6] Physical activity has numerous health benefits for older adults, helping reduce the risks of

diseases such as cancer, diabetes, cardiovascular disease and stroke and delaying the onset of dementia.[6] Being consistently active for over 6 months has also been associated with a reduced incidence of falls.[7] However, older adults do little exercise and have difficulties maintaining exercise over time.[6]

One technology of interest in supporting older people with meaningful activity, motivating exercise and encouraging social connectivity is virtual reality (VR). There have been increasing numbers of studies exploring VR use within care settings.[8] Here we report a study of a VR and omnidirectional treadmill system, MOTUS Adventure (MOTUS, formerly known as ROVR), designed for older adults.[9] The system is designed to motivate activity by physically navigating virtual environments that have meaning for the users. The use of environments with meaning for the user is essential to achieving the aim of meaningful activity, which, by definition, must be tailored to the user's preferences.[10]

The VR environments in MOTUS include places of cultural and historical interest, such as ancient forts, museums, nature scenes and cityscapes. These three-dimensional (3D) virtual environments represent places care home residents may no longer be able to visit physically. Access to culture (including heritage and history) is a human right,[11] with documented health and well-being benefits, including improved quality of life, opportunity for reminiscence and connectedness to spaces and history.[12] Older adults in particular are invested in heritage, yet physical access to sites of cultural or heritage interest is often limited due to site characteristics and conservation issues (eg, uneven ground, inability to implement accessibility features).[13] The Heritage Alliance annual report[12] recommended a partnership between heritage and health to overcome access challenges and help realise the full potential of heritage for health and well-being. The COVID-19 pandemic highlighted the potential for online access to culture and heritage in place of physical access,[14] with VR offering possible solutions.[15] The use of VR to visit cultural and heritage sites can therefore aid in inclusivity for those with physical and sensory barriers,[15] particularly prevalent among older adults. The opportunity for reminiscence brings additional well-being potential, with reminiscence thought to provide stimulation, enjoyment, a sense of self-worth and achievement.[16] It should also be considered, however, that some concerns have been raised about the use of VR with older adults.[8] For example, older adults can have vision-related challenges, and motion sickness in VR specifically for older adults should be thoroughly explored in research with end-users. Future research should consider how to achieve the intended exercise intensity and access to heritage and culture while ensuring resident comfort over suitable durations of VR use.

Alongside potentially motivating physical activity and providing access to cultural and heritage sites, MOTUS allows for 'walks' to be socially interactive, with multiple users joining the same virtual environment to 'walk and talk' together, regardless of their location. The aim of the system, therefore, is to enhance social connectedness and improve physical health through meaningful activity.[17] However, prior to this study, MOTUS had never been used in care homes. With any technology, there may be significant differences in opinions between developers and end-users,[18] so older people's feedback on the design of technology is essential. Given potential safety concerns and the need to understand how to implement the technology, we first needed to explore care home staff opinions on feasibility, design and potential for older care home residents.

## METHODS

### Aims

The aims of this study were to explore staff perceptions towards the acceptability and feasibility of the MOTUS Adventure system for older adult care home residents. An understanding of whether staff believe the technology will be feasible, acceptable or useful for care home residents is essential ahead of any direct feasibility study or implementation evaluation. Following study completion, results from the analysis were provided to the technology developer to help design iterations based on staff feedback on potential improvements and design requirements.

### Design

This study was a qualitative study based on technology interactions and focus group interviews, allowing staff to try and then discuss the equipment. The results were presented to the technology developer, allowing them to modify the design. The results were presented verbally during meetings and in written form later, based on a full analysis. We interviewed the Chief Executive about this and their experience of business working with research. This manuscript has been constructed following the Consolidated criteria for Reporting Qualitative research checklist for qualitative research.[19]

### Location and dates

Data collection on perceptions of the MOTUS standing treadmill and VR system took place from September to November 2021 in Looe, Liskeard, Saltash, St Austell, Perranporth, Camborne and Truro in Cornwall, UK. Feedback (from the participants and our interactions) was provided to the developers at the time through verbal meetings and in written format via email. A follow-up interview with the technology developer took place via Zoom in September 2022.

### Materials

The MOTUS Adventure omnidirectional treadmill with a Pico Neo 2 VR headset to immerse the users in virtual environments, is shown in figure 1. The headset is equipped with a 4K LCD display that provides a resolution of 1920×2160 pixels per eye with a field view of 101 degrees. The treadmill is a smooth plastic dish coated thinly with

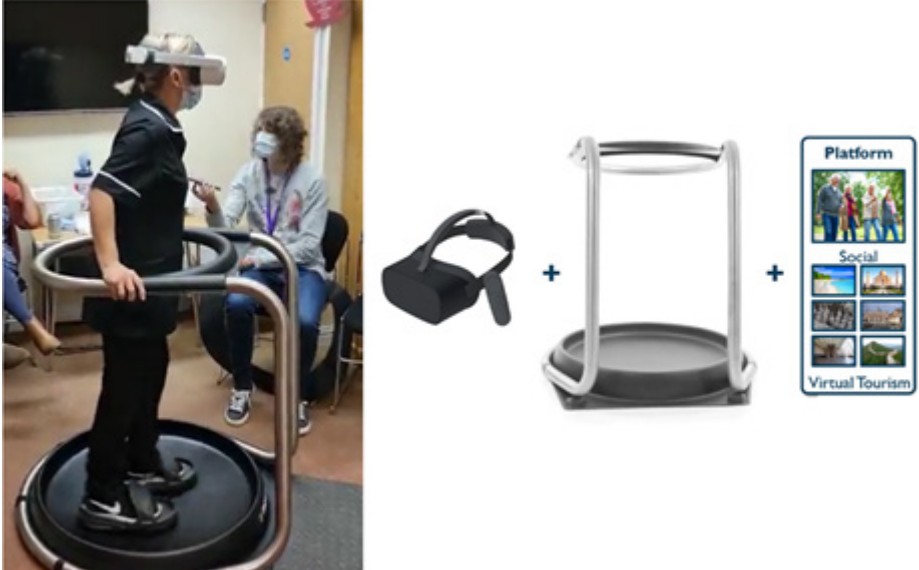

**Figure 1** (Left) Care home staff member interacting with the MOTUS Adventure treadmill with the researcher recording initial reactions. (Right) VR headset with the MOTUS Adventure omnidirectional treadmill and an example of the app to access VR worlds. VR, virtual reality.

lubricant, on which people stand wearing overshoes with ceramic plates on the base. This allows for a low friction, slippery surface on which people 'slide' their feet backwards and forwards to represent walking. The treadmill links to the MOTUS PC software (in this study, we ran the programme on a Lenovo ThinkPad L13) and VR system (figure 1) via Bluetooth, which together translate movement on the treadmill into movement through VR worlds. To change direction, participants swivel around and point their bodies in the direction of their desired travel. VR worlds include 3D-scanned, real-world, heritage locations as well as animated worlds ranging from museums to simple houses (figure 2). The MOTUS VR worlds can also be social experiences, with several users entering the same world from distant locations and walking and talking together. Within the workshops, participants were first offered the opportunity to walk in the simple, animated house to become accustomed to the experience before

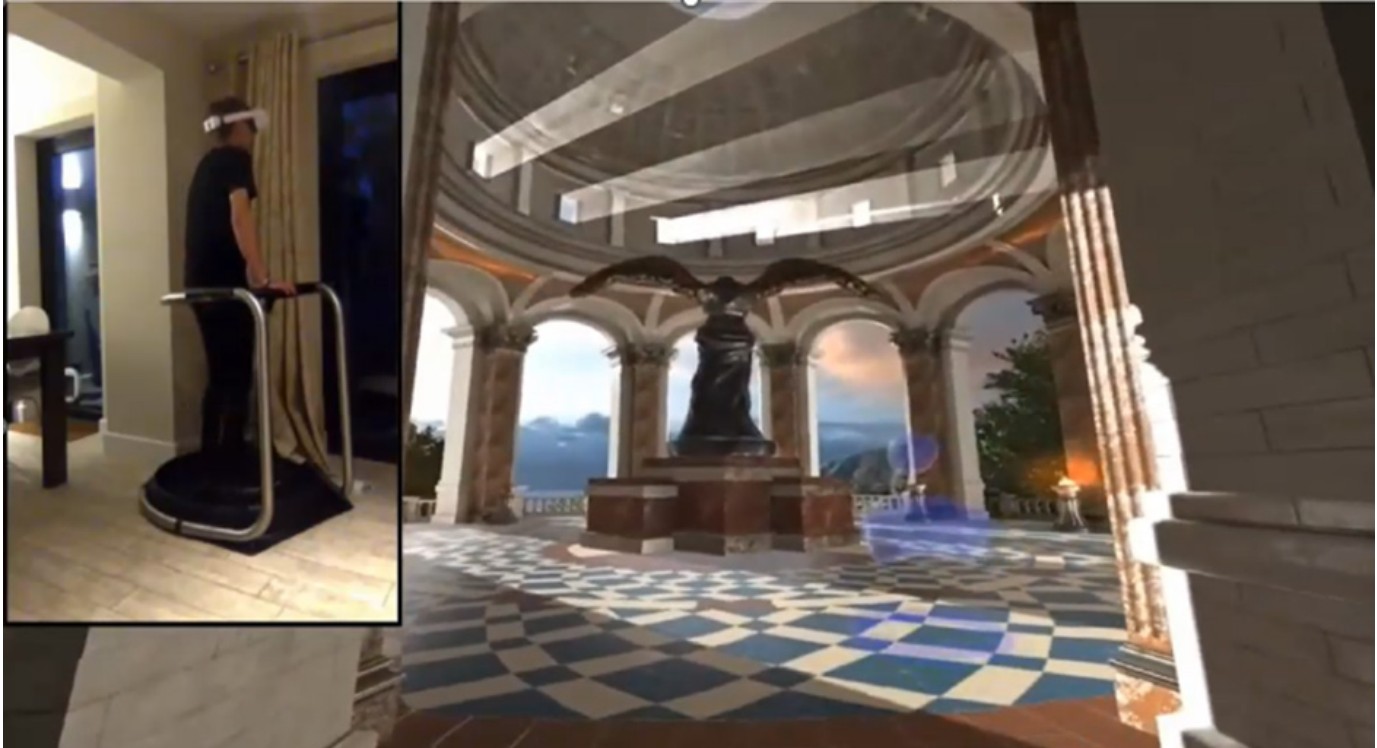

**Figure 2** Example of the virtual museum participants could explore by walking on the MOTUS Adventure.

trying more complex models such as heritage locations or museums. Each participant tried at least two or three of the VR environments. The participant was always 'joined' in the VR world by a researcher (as an avatar) linking into the social world through a computer and walking with them via keyboard controls to allow demonstration of the social angle.

## Procedure

### Recruitment

Convenience sampling was used. The health professionals were recruited at the University of Plymouth e-Health events through an approach by the researcher (HLB). Health professionals read participant information sheets and signed written consent forms before engaging in the workshop. All health professionals had experience with older adult care in various forms, some with movement/exercise in particular (care homes, physiotherapy and occupational therapy). The care homes were approached by email by HLB, chosen based on prior collaboration with the researcher, to scope interest in trying the MOTUS. For care home contacts who were interested, researcher visits were arranged and participant information sheets and consent forms were provided. It is important to consider the impact of prior collaboration on results; although the participants may have had enhanced exposure to other technologies through prior work, they had not previously been involved in any VR studies.

### Workshops

Two researchers (HLB and KJE, both PhD holders with a background in digital health research and a specific interest in social care) engaged in eight interaction workshops where a total of 44 key stakeholders (health and care staff) were provided with demonstrations of the MOTUS Adventure and the opportunity to try the technology themselves. All staff tried the MOTUS having at least 5 min of interaction, although some interactions were longer (>15 min), and all staff watched as other staff used the technology, including via casting of the VR world onto a computer screen, to help further inform their opinions through increased exposure to the technology in use. We used audio recording devices to capture comments during use. All 44 staff then participated in focus group discussions to share their perceptions. These were also audio-recorded. Sessions lasted from 43 to 110 (mean 71) minutes. Sessions had between three and eight participants (table 1). We were unable to link participants to distant MOTUS users as a social experience due to feasibility but we did demonstrate two users in one world (the researcher on a laptop and the participant on a treadmill) to facilitate a discussion around the potential social experience. One of the workshops engaged health professionals such as occupational therapists and physiotherapists, and the other seven workshops engaged care home staff and managers.

For the one follow-up interview (approximately 10 months after the data collection and feedback to the business based on participant perceptions), the technology developer was invited to take part in an informal interview to discuss their perceptions of the impact of the academic feedback from the focus groups on their products and processes. The interview was conducted by a separate researcher (ACW) from those who had been collaborating with the technology developer for the purpose of this study, to avoid bias. The interview was 1 hour in duration.

## Ethics

This study received ethical approval from the Faculty of Health Ethics Committee at the University of Plymouth, project ID 2887 (August 2021). All participants gave written informed consent.

## Patient and public involvement

Considering the timing of this study, with health and care staff capacity still limited due to the ongoing effects of the COVID-19 pandemic, it was not possible to involve them in the design, reporting or dissemination of this research.

| Table 1 | Participants at each interaction workshop |
|---|---|
| **Site** | **Participants (n)** |
| Workshop at University site 1 | General health and care professionals, including occupational therapist, physiotherapist and patient involvement expert |
| Workshop at University site 2 | Supported living manager and care home staff |
| Workshop at care home, 1-bed to 26-bed residential care home | Care home manager, compliance manager, activity coordinator and carer |
| Workshop at care home, 2-bed to 83-bed nursing home with unit for people with dementia | Care home compliance manager, clinical needs manager, activity coordinator and carers |
| Workshop at care home, 3-bed to 47-bed care home for people with dementia | Care home manager and carers |
| Workshop at care home, 4-bed to 33-bed care home | Care home manager and carers |
| Workshop at care home, 5-bed to 40-bed care home | Care home manager and carers |
| Workshop at care home, 6-bed to 60-bed nursing home for people with dementia | Residential care nursing manager and carers |

### Data collection

Stakeholders provided feedback during their interaction with the MOTUS Adventure treadmill and following the interaction during informal focus groups (interview schedule available in online supplemental appendix A). We used a semi-structured interview approach, following the schedule but deviating from natural conversation and follow-up questions on interesting comments. The feedback was audio-recorded and transcribed verbatim. The interview with the technology developer was also audio recorded and transcribed verbatim; there was no interview schedule used, but general queries were posed to the developer on their perceptions towards the feedback from this study.

### Data analysis

Researchers (HLB, LC, RB, SAT, JR, ACW, TK, SG and AVR) analysed the transcripts using thematic analysis. Braun *et al*[20] detailed the process of thematic analysis, a form of analysing qualitative data that is often used in the health and social sciences. As prescribed, the researchers first became familiar with the data set (through reading and understanding the full transcripts) before generating initial codes by rereading and looking for meaning, searching for themes across the codes from all transcripts and reviewing and defining those themes. The subtype of thematic analysis used was reflexive, in generating codes from explicit content, which evolves and adapts through considerable analytic work, to produce themes, representing an understanding of meaning across a dataset.[20] Analysis was inductive, as researchers had predetermined aims and therefore were seeking an understanding of participant perceptions towards the device among the data. Analysis was conducted by nine researchers due to the size of the dataset, with a considerable amount of qualitative data collected. All nine researchers conducted initial coding, with transcripts divided between researchers; then HLB, LC and RB led iterations of reviewing and defining themes; however, the whole research team had two meetings to explore different perspectives on the data and codes to enhance the validity of interpretation through perspectives from multidisciplinary expertise (occupational therapy, psychology, physical activity science, computer science, social care research and digital design). The range of experience provided ensured broad perspectives were considered in our interpretations. We used the NVivo 12 data analysis software for data management to ensure a clear audit trail, enhancing dependability.[21] The involvement of a multidisciplinary team with various expertise in the data analysis processes enhances the credibility of our results.[22] The researchers who conducted the analysis had diverse (yet related to the topic) disciplinary backgrounds (including engineering, health and design), increasing the robustness and significance of the results (all researchers included have or are working towards a PhD and strong experience in data analysis). The process for the analysis of the follow-up interview with the technology developer was the same, although conducted by GCR, ACW and HLB.

### Participants

In total, across the eight interaction workshops, 44 participants interacted with the MOTUS systems VR treadmill and provided evaluative feedback (table 1). The participants included a broad range of appropriate stakeholders for English care homes, including across all levels of the operational care home staff and additionally various health professionals with expertise relevant to movement and activity (eg, occupational therapists and physiotherapists). For the follow-up interview with the technology developer, there was one participant, the chief operating officer of the MOTUS systems technology developer.

## RESULTS
### Section 1: Workshop results

Here we discuss the results of the eight interactive workshops exploring stakeholder perceptions towards the VR-motivated MOTUS treadmill. In section 1, quotes have been identified by the site where the data were collected rather than by individual participants. The roadshow/workshop nature of data collection and recording meant identifying individuals within busy transcripts was not feasible. In section 2, we share the results of the follow-up interview with the technology developer. The themes and codes identified in the analysis of the eight workshops are shown in table 2.

### Anticipated benefits

Participants noted a range of perceived benefits of using MOTUS, with many focusing on the potential for physical benefits, such as providing '*good exercise*' (Care Home 2) and being '*particularly […] fantastic for rehab*' (Care Home 5). Participants noted that finding motivation to complete standard exercises can be a '*struggle*' and '*boring*' (Care Home 5), while the VR content would mean '*they can actually walk on that, something to motivate them to do it*' (Care Home 5).

Participants also noted the potential enhanced accessibility of '*tourist sites*' (Care Home 5), enhancing visiting potential, particularly for those with '*reductions in mobility*' (Care Home 1). One staff member reported '*some (residents) are just not able to get out anymore […] you're providing the opportunity to see the place that they love*' (Care Home 4). This experience was additionally expected to create '*enhanced communication*' (Care Home 5), prompting conversations between residents with each other and the staff. Care staff additionally felt that reminiscence could be an additional benefit as it may '*trigger memories*' for the residents (Care Home 2).

### Acceptability

Further to the range of potential benefits noted by care staff, participants generally demonstrated good acceptability towards the device, often praising the technology

**Table 2** The results of thematic analysis of the eight interaction workshops

| Theme | Initial codes, *subcodes* |
|---|---|
| Anticipated benefits | Virtual visits; interesting local sites; virtual tourist sites |
| | Physical activity/exercise; motivating exercise |
| | Prevents boredom |
| | Rehabilitation |
| | Staff use |
| | Social interaction |
| | Well-being |
| | Accessible experiences; different to care home |
| | Stimulates the brain |
| | Restorative |
| | Reminiscence and reminiscence benefits |
| | Expectations and challenges positively |
| | Positive distraction and novelty |
| Acceptability | Positive comments |
| | MOTUS is interesting |
| | Adoption desired |
| | Ease of use: headset initially heavy but comfortable, physically easy, treadmill is easy to enter, treadmill is easy to move on and VR is novel experience |
| Concerns of use | Secureness |
| | Safety concerns: hitting objects, shoes, safety hazards and danger |
| | Lack of control: movement issues |
| | Risk of falls/dangerous: history of falls, could cause falls, risk of injury—hips |
| | Accessibility |
| | Prefer people go out |
| | Heavy shoes |
| | Lack of motivation |
| | Uncomfortable movement |
| | Bigger environment more physically demanding |
| | Critique |
| | Potential issues |
| Concerns of negative effects | Side effects: encouraging shuffling/sliding and unnatural movement |
| | Dizziness |
| | Disorientating: orienting to environment, coordination between VR and feet, height changes, motion and social angle confusing |
| | Concentrating more on the exercise |
| | Depth perception difficulties |
| | Motion sickness in VR |
| | Fear |
| Suitability/ Unsuitability | Extra support: supervised use required, guidance and guided navigation, human resources, number of residents, differences between residents and residents unable to use |
| | Diagnosis: balance, dementia/cognition, fear of falling, have to be able, incontinence, influence of medication, limited mobility, problems with vision, neck pain, neuromuscular, lived problem and vertigo |
| | Suitability for residents: 99% not suitable, suitable for residents 20 years younger, would not recommend to people with mobility issues |
| | Ethics |
| | Risk |
| | Size of technology: WiFi connectivity |
| | Familiarisation: adapting to use, unfamiliar, adjusting to headset and familiarity |

Continued

**Table 2** Continued

| Theme | Initial codes, *subcodes* |
|---|---|
| Improvements | Content suggestions: cartoon-like images and new environments |
| | Inclusivity: adaptable bar height, fitting to a wheelchair, people less mobile, access could be improved and physical requirements |
| | Safety improvements: softer impact bars, stop slipping, awfully slippery and difficulty exiting treadmill |
| | Add arm movements |
| | Cast visual for group to watch |
| | Seated version: square, headset use only and adaptations |
| | Possibilities/exercise bike |
| Current design | Good content: visual features and spatial features |
| | Liked headset |
| | Safety features |
| | Technology issues: glitching, loading worlds and speed of movement |
| | Ergonomics |
| | Mechanics |
| | Resolution |

Evidence to support each code can be found in the online supplemental appendix B and is synthesised in the narrative below.
VR, virtual reality.

as '*amazing, absolutely amazing*' (Care Home 3), '*honestly, I think it's amazing*' (Care Home 6). One care home manager even reported that she would '*buy it for the staff and residents […] so that we actually had one permanently*' (Care Home 5).

## Concerns of use
Despite the general acceptability and praise towards the concept and appreciation for its potential benefits, participants noted some concerns about the suitability of such devices for older adults and the care setting. Most concerns relate to the safety of vulnerable older adults using a low-friction stand-up treadmill. One care home manager reported that for care home residents specifically, the design was '*unsafe*' (Care Home 4), with another suggesting it was '*a danger*' due to being '*slippery*' (Care Home 5). The worry was that generally care home residents are '*not as agile as they used to be*' and could '*potentially fall*' (Care Home 3). This could result in injuries to '*ribs […] or your arm, break an arm*' (Care Home 5), with one carer suggesting, '*I can see a few hips going*' (Care Home 2). As a result of these concerns, participants felt they would not want residents using the treadmill '*without a staff member*' (Workshop 2).

## Concerns of negative effects
Beyond some safety concerns of residents using a stand-up, slippery treadmill, broader concerns were raised around '*motion sickness*' (Care Home 3), feeling '*dizzy*' (Care Home 5) and unnatural movements '*because you're not stepping*' (Care Home 5), '*it'll feel a bit weird*' (Care Home 6).

## Suitability/unsuitability
Participants debated among the older adults in their care who MOTUS would and would not be best suited

to. Based on the concerns presented above, care home staff echoed that residents would '*need to have much more support*' to use the treadmill (Care Home 3), and to be '*supervised*' (Care Home 2), which would mean '*we're going to have to get the staff on board*' (Workshop 2). The treadmill was felt to be more suitable for residents with '*more stability when they're on their feet*' (Care Home 5), while care staff listed diagnoses that would make the standing treadmill less suitable. Suitability concerns were raised for people with '*lots of medication (that) affects their balance*' (Care Home 1), people with '*dementia*' (Care Home 4), who could '*totally freak*' (Care Home 6) and people with '*vertigo*' (Care Home 1). Despite the concerns, participants acknowledged that '*just like anything else, it's time to get used to it*' (Care Home 2), suggesting that prolonged use may encourage more confidence.

## Improvements
Although participants were generally accepting and appreciative of the concept of VR to motivate physical activity and acknowledged the potential benefits, the stakeholders were conscious of potential safety concerns, which led to numerous participants suggesting a redesign of the treadmill to a '*seated unit*' which would '*be more beneficial to our client base*' (Care Home 2). Care staff suggested less mobile residents could be '*sat using their feet*' (Care Home 4) on a flat platform, to navigate the VR worlds without the risk of the standing, low-friction treadmill. Participants felt this would improve the inclusivity of the experience as '*most (residents) come in in wheelchairs*' (Care Home 1). With these adaptations in mind, participants also shared thoughts on desired content for inclusion as a VR world, such as '*gardens*' (Care Home 4), '*under the sea*' (Care Home 5) or '*beaches*' (Care Home 4).

**Table 3** The results of thematic analysis of the follow-up interview with the technology developer

| Theme | Initial codes, subcodes |
|---|---|
| Benefits | Beneficial outcomes: physical benefits, social benefits, mental benefits, pushing personal boundaries, benefits of working with the University |
| Process of development | Technology development: seated version considering inclusivity: age, care homes, medical issues, different sectors phases of product development |
| Challenges | Cost and resources |

## Current design

While care stakeholders noted required adaptations to use VR to motivate movement and exercise safely for the target population of older adults and care home residents, participants praised aspects of the current design, including the VR content itself, as '*amazing, brilliant, it is amazing, that is phenomenal*' (Care Home 5). They felt the headset '*doesn't feel too heavy on my head*' (Care Home 5), but '*feels really comfortable*' (Care Home 4). Participants did also note some technical glitches to be overcome, such as not '*going anywhere*' (Care Home 5) when walking and '*flashing all over*' (Care Home 5). These forms of issues were reported to the developers for bug fixes during the study period.

## Section 2: Results of the follow-up interview with the technology developer

Quotes have not been provided with an identifier, as only one participant took part in this follow-up interview. The main themes, initial codes and subcodes resulting from the analysis are provided in table 3.

## Benefits

The follow-up interview with the technology developer highlighted a range of benefits born from the industry–academia collaboration. This included an initial code directly relating to the *benefits of working with the University as* the participant reported it was '*really helpful*' through the opportunity to '*apply the minimum viable product (and figure) out where it works and where it doesn't*' stating '*you get really fast feedback*'. The developer felt academic collaboration and real-world feedback from stakeholders provided '*feedback we can work with*'. The participant stated: '*it's really important for commerce (industry) to understand this, they need to work with people who are already experts*' as the participant felt universities '*have the freedom to do what most companies can't*'. In this case, the technology developer benefited from the extensive experience the researchers had in working within social care and with older adults, '*you were working in this environment and already had that knowledge and understood the challenges involved*'. The developers felt bringing their specific technical knowledge and combining it with the university's evaluation expertise meant '*we didn't have to […] learn it ourselves […] or teach you […], all we had to do*

*was provide the equipment and gain feedback*'. The developer felt having evaluation through the University provided a '*helpful*', '*independent view*' '*which is unbiased*'.

The technology developer also noted the benefits they perceived of the technology itself, based on the interactions during the study period and subsequent collaborative studies (to be published elsewhere). The developer discussed how older adults, in general, can need '*motivation to exercise*' with one particular challenge '*in the Northern hemisphere*' being '*winters (which are) long and wet and dark*' leading to older adults becoming '*deconditioned over the winter*'. Technologies such as MOTUS provide an indoor solution to motivate physical activity, which could be achieved '*on a daily basis*'. The developer also shared the benefits of the potential social connection facilitated within the MOTUS VR worlds, '*they (older adult) visited (cultural location in VR) before, they'd love to visit. They can meet up with friends, they don't have to be in the same location, they can join, without necessarily having VR equipment yeah. And that makes this a social interaction*'. Beyond the physical activity and potential for social contact, the developer felt they observed that the technology '*cognitively de-stressed*' older adults, meaning a range of potential benefits were observed.

## Process of development

During the interview, the technology developer detailed how feedback from the current study helped lead to the development of a seated version of the MOTUS treadmill (MOTUS Explore), which they felt was particularly required for '*the level of condition we found in care homes*' as '*they're really seated […] without major assistance for a long time*' meaning they couldn't use the standing MOTUS. The participant stated: '*one of the limitations, […] is the inability to move around safely*'. The developer responded with '*a plate, which could be sat in front of people if they were seated and enable them to move*'. The participant stated: '*it was a real opportunity to include people who would otherwise be excluded*', demonstrating significant value in this insight and product development as a result of the collaboration.

## Challenges

The technology developer also shared some thoughts on challenges they face generally, such as cost. Before 2012, VR development was only for those '*with a million pound budget*', and though costs have fallen dramatically the feeling is that '*costs of development are still pretty high at present*' with '*various challenges around the world adding to those costs*'. Development costs and associated challenges further highlight the value of evaluation, iterative design and co-development with stakeholders, ensuring early developments are suitable and desirable for the intended audience before funds are wasted on inappropriate mass production.

## DISCUSSION

This study explored health and care staff perceptions towards the acceptability, feasibility and potential of VR-motivated treadmill exercise for older adult care home residents, which is important due to the impact

of the ageing population and the growing burden of disease.[2] Research into the potential for technological innovations to support healthy ageing is essential to help respond to these challenges.[3] Our study adds to the literature on the design of technology to encourage and support physical activity and exercise in older adults and care home residents.

Novel technologies such as MOTUS have the potential to support healthy ageing. Our results demonstrate clear promise for such technologies to motivate and sustain increased physical activity in older adults and people living in care homes, with our large sample of care staff generally agreeing on the potential benefits of exercise and activity. Implementation of products such as these would likely have benefits for meaningful activity, meeting the NICE[10] definition of providing an activity that promotes health and well-being through an activity that has meaning to the individual and is tailored to their preferences. Our results have demonstrated an interest and preference among stakeholders for cultural, heritage and natural sites for use as a meaningful VR experience. Further work is now required with older adults themselves to confirm their preference for such an activity.

Staff also perceived benefits for social interaction, both through the social aspect of the VR platform and additionally through increases in residents talking to each other, their families and carers, with the technology experience providing a focal point. This could have implications for the experience of loneliness among the target population.[5 23] Our results suggest the technology was also expected to allow for reminiscence and enhanced experiences by allowing older adults and care home residents to 'visit' places and heritage sites they could no longer access. Staff also noted that older adults may be able to 'visit' places significant to them through memories from the past. The benefits of access to heritage locations are well documented[12]; therefore, future research could explore and document the impact virtual access that is paired with technology that involves exercise to move around heritage sites has on health and well-being outcomes. Further exploration is also required to quantitatively assess the impact of physical activity (or motivation to exercise) over longitudinal periods on outcomes for older adults. Additional future research should also seek to document the technical competence required for care staff to implement and facilitate such technologies without researchers present.

However, despite the positive perceptions towards the concept of VR to motivate activity and the use of some form of treadmill to encourage movement and muscle use, care staff raised concerns around product design, mostly related to safety. Participants were concerned that for the frailer older adults and care home residents, standing on the low-friction treadmill could present a fall risk. When the results were provided to the technology developer, this study had immediate implications for product design. Based on this stakeholder feedback, the MOTUS Systems team subsequently developed the Explore system, a seated version of the omnidirectional treadmill that would not present a fall risk while still motivating activity and lower leg movement to build up leg strength. The seated Explore system is currently undergoing evaluation with stakeholders as part of the ongoing Generating Older Active Lives Digitally project.[24] Beyond this immediate effect, this study has broader implications for those designing physical activity technologies aimed at supporting elderly care. It demonstrates that care stakeholders are open and accepting of such technologies, believing the concept to be positive, while also providing useful design, hardware, and content insight for future developments. This value is clearly demonstrated in this study through the follow-up interview with the developer and subsequent new product development.

This study also reinforces previous work,[1 18] showing the importance of engaging stakeholders in the design and testing of technologies aimed at health and care. The follow-up technology developer interview showed the value of academic-industry collaboration. The developers were able to use these results to adapt the technology to a seated treadmill, resulting in an enhanced safety experience for those unable to comfortably stand and increasing the accessibility of the experience and its potential health and well-being benefits.

### Strengths and limitations
Our study achieved a relatively large sample for qualitative work and gained deep and thorough insight into care staff perceptions. Our analysis used rigorous methods, and the topic of enquiry was novel in exploring VR-motivated physical activity specifically for older adult care home residents.

The strong qualitative approach produced data that was substantial and complex, and although the themes and categories presented here encompass all data, there is additional evidence of more profound and complex forms of engagement with VR technologies, such as speculative feedback or feelings of telepresence. These additional dimensions warrant further investigation.

There were two limitations. First, the duration for which participants interacted with MOTUS was short. Our results, therefore, may be over-optimistic and influenced by the novelty effect. Longitudinal work would have better helped identify barriers and facilitators to use that would not be immediately recognised. The concerns about negative effects reported in our results, such as dizziness and motion sickness, would likely have been better assessed in longitudinal work, allowing time for participants to familiarise and overcome any initial negative sensations. These types of reactions are common among all VR systems, with work underway to generally reduce cybersickness experiences for users of head-mounted displays.[25] Research has shown that repeated exposure to the same VR content will reduce the severity of motion sickness[25]; thus, a longitudinal or repeated measures study would more accurately report on the degree of barrier presented by the unpleasant sensations. As with the potential reduction in

motion sickness over time, participants themselves noted that it would take time to gain confidence in using the MOTUS device with older adults and care home residents, therefore demonstrating the potential to be more accepting over time.

Second, we only sought staff opinions on the potential future use of these devices, rather than older adults themselves. However, for devices such as these to be used by older adult care home residents, as intended by the developers, they first need to be accepted and adopted by the staff team, who would be responsible for buying and facilitating use. To overcome both the limitation of a short interaction period and staff stakeholders only, we have conducted a 6-week implementation study with the MOTUS device to be reported separately. It should also be considered that we did not explore device cost or site investment potential; therefore, for real-world implementations, cost would be a consideration.

## Conclusion

The use of VR-motivated exercise on an omnidirectional treadmill holds potential based on the perceptions of care staff for increasing exercise of older care home residents, while also providing virtual access to heritage sites, encouraging reminiscence and social connectivity, and promoting participation in a meaningful activity. A standing device, however, was not appropriate for a care home population, and a new design is now being used for research with care home residents. User-centred design of health and care technologies is essential, and this study has demonstrated significant benefits for industry partners in collaborating with university institutions for independent evaluation with end-users.

**Author affiliations**
[1]Centre for Health Technology, University of Plymouth, Plymouth, UK
[2]Faculty of Health Sciences and Sport, University of Stirling, Stirling, UK
[3]Faculty of Arts & Humanities, University of Stirling, Stirling, UK
[4]Department of Applied IT, University of Gothenburg, Gothenburg, Sweden
[5]Faculty of Arts, Humanities & Business, University of Plymouth, Plymouth, UK
[6]Institute of Cardiovascular and Medical Sciences, University of Stirling, Stirling, UK
[7]Faculty of Medicine, University of Leeds, Leeds, UK
[8]Faculty of Social Sciences, University of Stirling, Stirling, UK

**Acknowledgements** We would also like to thank our collaborating care homes for hosting our workshops and colleagues on the EPIC project who helped establish the collaboration.

**Collaborators** On behalf of the GOALD project: Catherine H. Hennessy, Faculty of Social Sciences, University of Stirling; Ray B. Jones, Faculty of Health, University of Plymouth; Richard Haynes, Faculty of Arts & Humanities, University of Stirling; Anna C. Whittaker, Faculty of Health Sciences & Sport, University of Stirling (Core Management Team); Sheena Asthana, Plymouth Institute for Health and Care Research (PIHR), University of Plymouth; Rory Baxter, Faculty of Health, University of Plymouth; Hannah Bradwell, Faculty of Health, University of Plymouth; Arunangsu Chatterjee, Faculty of Medicine and Health, University of Leeds; Pete Coffee, School of Social Sciences, Heriot-Watt University; Leonie Cooper, Faculty

of Health, University of Plymouth; Alison Dawson, Faculty of Social Sciences, University of Stirling; Katie Edwards, Faculty of Health, University of Plymouth; Swen Gaudl, Faculty of Science & Engineering, University of Plymouth; Tanja Krizaj, Faculty of Health, University of Plymouth; Gregory Mannion, Faculty of Social Sciences, University of Stirling; Gemma Ryde, Institute of Cardiovascular and Medical Sciences, University of Glasgow; Alejandro Veliz Reyes, Faculty of Arts, Humanities and Business, University of Plymouth; John Ritchie, Faculty of Arts & Humanities, University of Stirling; Simone Tomaz, Faculty of Health Sciences & Sport, University of Stirling; Alison Warren, Faculty of Health, University of Plymouth; Karen Watchman, Faculty of Health Sciences & Sport, University of Stirling; Katherine Willis, Faculty of Arts, Humanities and Business, University of Plymouth, and partners—Active Stirling Ltd., Generations Working Together, Hearing Loss Cornwall, Nudge Community Builders, South Asian Society (Devon and Cornwall), Sporting Heritage, Sports Heritage Scotland, St. Breward Community, iSightCornwall, UKActive.

**Contributors** HLB and KJE led the design of the research. HLB, KJE, LC, AW and RB conducted data collection. HLB, LC, RB, AW, TK, SAT, JR, GCR, SG and AVR conducted data analysis. RBJ oversaw the project design, completion and substantively contributed towards the write-up, which was led by HLB. SA, KW, CHH, ACW, AC and RH oversaw study design and completion as GOALD senior management with RBJ and reviewed and contributed to the manuscript. All authors reviewed, edited and agreed to the final manuscript. HLB is the author acting as guarantor.

**Funding** This work was supported by UKRI Healthy Ageing Social, Behavioural and Design Research Programme grant number ES/V016113/1 as the project was completed by researchers on the GOALD project and the project also purchased MOTUS equipment for the study, with further support from the EPIC (eHealth Productivity and Innovation in Cornwall and the Isle of Scilly) project which was part funded by the European Regional Development Fund. Additional funding for the EPIC project was received from University of Plymouth. ERDF Grant number [05R18P02814], as EPIC also purchased some of the MOTUS equipment used. Finally, support was also provided by Innovate grant 10004423, which funded development, equipment and support provided by MOTUS Systems Ltd.

**Competing interests** None declared.

**Patient and public involvement** Patients and/or the public were not involved in the design, or conduct, or reporting, or dissemination plans of this research.

**Patient consent for publication** Consent obtained directly from patient(s).

**Ethics approval** This study involves human participants and was approved by the Faculty of Health Ethics Committee at the University of Plymouth, project ID 2887 (August 2021). All participants gave written, informed consent. Participants gave informed consent to participate in the study before taking part.

**Provenance and peer review** Not commissioned; externally peer reviewed.

**Data availability statement** Data are available upon reasonable request. Data are available upon reasonable request to the corresponding author.

**ORCID iDs**
Hannah Louise Bradwell http://orcid.org/0000-0002-9103-1069
Katie Jane Edwards http://orcid.org/0000-0001-6212-6010
Anna C Whittaker http://orcid.org/0000-0002-5461-0598
Ray B Jones http://orcid.org/0000-0002-2963-3421

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
