## [Reviewer comments · BMJ Open]

ARTICLE DETAILS

TITLE (PROVISIONAL)	Staff perceptions towards virtual reality motivated treadmill exercise for care home residents: A qualitative feedback study with key stakeholders and follow-up interview with technology developer.
AUTHORS	Bradwell, Hannah; Cooper, Leonie; Edwards, Katie; Baxter, Rory; Tomaz, Simone; Ritchie, John; Gaudl, Swen; Veliz-Reyes, Alejandro; Ryde, Gemma; Križaj, Tanja; Warren, Alison; Chatterjee, Arunangsu; Willis, Katharine; Haynes, Richard; Hennessy, Catherine; Whittaker, Anna; Asthana, Sheena; Jones, Ray

VERSION 1 – REVIEW

REVIEWER	Gjesdal, Beate Eltarvåg Western Norway University of Applied Sciences
REVIEW RETURNED	29-May-2023

GENERAL COMMENTS	Background: The third section of the manuscript explores the cultural and historical aspects of virtual reality (VR) and highlighting its benefits. However, one important aspect that could be further addressed is the issue of motion sickness. Considering that older adults may have vision-related problems, it would be valuable to discuss whether VR poses any challenges in this regard. The last section effectively summarizes the potential advantages of ROVR, particularly in terms of promoting physical activity, providing access to cultural and heritage sites, and facilitating social interaction. This section aligns well with the ICF terminology of body structure and function and participation, making it particularly engaging and noteworthy in my opinion. Methods: Aim If the study were to explore feasibility of the ROVR adventure system, why isn't this a feasibility study? The aims of this study appear to be twofold, and the second aim related to technology development is somewhat difficult to comprehend. Design Is it a qualitative feedback study or a qualitative study based on focus group interviews? Were all the findings presented to the developer, or only relevant findings? And how does this part of the study fit in to the "main study"? Further clarification on the nature of the study design and the interrelation between the two aims would greatly enhance the understanding of the research. Materials
--

	While this section contains a wealth of technical information that would be beneficial for individuals interested in testing the system, it may not be readily accessible to all readers. However, based on the background section, there was an intriguing discussion regarding the potential of physical activity and motivating exercises. It would be helpful to know more about the content covered during the workshop to gain a better understanding of how these concepts were explored in practice. P4 S31-34: How was the ROVR VR world used in the workshop? Procedure In this section, there is a combination of information regarding participant recruitment and a partial description of the workshops. It might be beneficial to provide a more comprehensive overview of the workshops in the earlier section. Additionally, it is unclear how a minimum interaction time of five minutes is deemed sufficient to answer the research question. Further elaboration on this point would enhance understanding. Regarding participant recruitment, it is important to consider how the results may be influenced when recruiting individuals with prior collaborations. Very good to have a separate researcher conducting the developer interview regarding bias. However, in the focus groups, the duration of the interviews is mentioned, and it may be beneficial to include the interview duration in this section as well for consistency and additional context. Data collection When was the finding presented to the technology developer, prior to or during the interview? Data analysis P5 S45-53: This is a general description on the analysis. Did the nine researchers perform all steps? How is the unit of meaning defined? HB, LC and RB led on iterations of reviewing and defining themes. However, it is unclear how the reliability of the codes (57 initial codes) was ensured in this study with multiple researchers analyzing the data. Were the codes mutually exclusive? Were all themes covered in all workshops? Since this research follows an inductive approach, did the interview guide undergo any modifications during the data collection process? Participants This might fit better earlier in the methods section. Results (or findings?) Section 1: Workshop results - In Table 1, only two of the eight sites are described as Workshop. P6 S50-57. The introduction of Section 1 and 2 might fit better under the Results heading than the Section 1 heading. Table 2 – This can be part of the methods section and referred to in the Data analysis. Section 2: Results of the follow-up interview with the technology developer There is a significant amount of space dedicated to the results from Part 2, which is not actually mentioned in the study's title or included in the objectives stated in the abstract. Discussion The results are divided into two sections, and it would be beneficial to structure the discussion accordingly. Further
--	--

	exploration is warranted on the results pertaining to physical activity and exercises for older adults, as well as how staff members can effectively implement these interventions. Additionally, it would be valuable to delve into the level of technical competence required for staff to successfully conduct such sessions with older adults. Strengths and limitations are thoroughly addressed in the study.
--	---

REVIEWER	Wang, Guodong Soochow University
REVIEW RETURNED	30-May-2023

GENERAL COMMENTS	This manuscript explores how to improve the physical activity of the elderly in nursing homes through the views and feedback of 44 healthcare professionals on ROVR. I think it is very interesting and practical. However, I have some suggestions for improving the quality of the manuscript that I hope the authors can seriously consider. Q1. In my experience, even healthy older people may wear a headset VR display for a bit longer (maybe more than 20-30 minutes?). Symptoms like dizziness or nausea are very noticeable. The number of elderly people suffering from chronic diseases in nursing homes is very large. In practice, how to achieve exercise intensity while keeping patients comfortable is a worthwhile challenge. Q2. I would like to know whether the author took into account the different feedback that may be caused by different levels of health management work environment and different subprofessional directions when conducting the survey of health management staff. In other words, can the 44 employees surveyed comprehensively cover the real thoughts of people in the same industry? Q3. The results are not clear due to the long content of the results. It is suggested that the authors simplify the relevant content to improve the readability of this manuscript.
---

VERSION 1 – AUTHOR RESPONSE

Reviewer 1:

Perceptions of staff towards virtual reality motivated omnidirectional treadmill exercise for older adult care home residents: A qualitative feedback study.

Title: Could benefit a more precise wording in the title.

Thank you for this comment, we've now made the title more concise.

Background: The third section of the manuscript explores the cultural and historical aspects of virtual reality (VR) and highlighting its benefits. However, one important aspect that could be further addressed is the issue of motion sickness. Considering that older adults may have vision-related problems, it would be valuable to discuss whether VR poses any challenges in this regard.

Thank you for this comment, we agree that motion sickness is an important consideration, particularly with older adults. We have added a sentence to BACKGROUND to this effect. In this paper we explore the perceptions of care staff rather than trial the VR with older adults themselves. We had already mentioned (in LIMITATIONS) the need for trials with older adults and that possible motion

sickness was something needing further exploration. We hope that by including this new sentence in BACKGROUND we have provided the relevant prompt for the later discussion of this aspect. Sentence added: *It should also be considered however, that some concerns have been raised on use of VR with older adults [8]. For example, older adults can have vision-related challenges, and motion sickness in VR specifically for older adults should be thoroughly explored in research with end-users. Future research should consider how to achieve the intended exercise intensity and access to heritage and culture while ensuring resident comfort over suitable durations of VR use.*

The last section effectively summarizes the potential advantages of ROVR, particularly in terms of promoting physical activity, providing access to cultural and heritage sites, and facilitating social interaction. This section aligns well with the ICF terminology of body structure and function and participation, making it particularly engaging and noteworthy in my opinion.

Thank you very much for this comment.

Methods: Aim If the study were to explore feasibility of the ROVR adventure system, why isn't this a feasibility study? The aims of this study appear to be twofold, and the second aim related to technology development is somewhat difficult to comprehend.

Thank you for noting this, we agree the previous wording was perhaps confusing. This was not a feasibility study, but instead exploring staff perceptions towards feasibility. We have amended the Aims section to clarify both the feasibility and technology development sentences as here: *The aims of this study were to explore staff perceptions towards acceptability and feasibility of the ROVR Adventure system for older adult care home residents, an understanding of whether staff believe the technology will be feasible, acceptable or useful for care home residents is essential ahead of any direct feasibility study or implementation evaluation. Following study completion, results from the analysis were provided to the technology developer to help design iterations, based on staff feedback on potential improvements and design requirements.*

Design Is it a qualitative feedback study or a qualitative study based on focus group interviews? Were all the findings presented to the developer, or only relevant findings? And how does this part of the study fit in to the "main study"? Further clarification on the nature of the study design and the interrelation between the two aims would greatly enhance the understanding of the research.

Thank you for this point, we have responded with the amendment to the design section as follows: *This study was a qualitative study based on technology interactions and focus group interviews, allowing staff to try, then to discuss, the equipment. Findings were presented to the technology developer allowing them to modify the design. Findings were presented verbally during meetings and in written form later, based on full analysis. We interviewed the Chief Executive about this and their experience of business working with research. This manuscript has been constructed following the COREQ checklist for qualitative research [19].*

Materials While this section contains a wealth of technical information that would be beneficial for individuals interested in testing the system, it may not be readily accessible to all readers.

We agree that some of the technical description is not very accessible for many readers of BMJ Open but feel that given the rapid evolution of these technologies it is important to have it in the text. We think it is easy enough to 'skip over' for readers keen to reach the next part of the paper.

However, based on the background section, there was an intriguing discussion regarding the potential of physical activity and motivating exercises. It would be helpful to know more about the content covered during the workshop to gain a better understanding of how these concepts were explored in practice. P4 S31-34: How was the ROVR VR world used in the workshop?

Thank you for this useful point, we have added the following to the materials session to help clarify: *Within the workshops, participants were first offered the opportunity to walk in the simple, animated house, to become accustomed to the experience before trying more complex models such as the heritage locations or museum. Each participant tried at least two or three of the VR environments. The participant was always 'joined' in the VR world by a researcher (as an avatar) linking in to the social world through a computer and walking with them via keyboard controls, to allow demonstration of the social angle.*

Procedure In this section, there is a combination of information regarding participant recruitment and a partial description of the workshops. It might be beneficial to provide a more comprehensive overview of the workshops in the earlier section.

Thank you, we believe this is now responded to with the above amendment.

Additionally, it is unclear how a minimum interaction time of five minutes is deemed sufficient to answer the research question. Further elaboration on this point would enhance understanding.

This is a useful point, thank you for noting, we have amended to improve the understanding as follows: *All staff tried the ROVR having at least 5 minutes of interaction, although some interactions were longer (>15 minutes), and all staff watched as other staff used the technology, including via casting of the VR world onto a computer screen, to help further inform their opinions through increased exposure to the technology in use.*

Regarding participant recruitment, it is important to consider how the results may be influenced when recruiting individuals with prior collaborations.

Thank you for noting this, it's an important consideration and should be acknowledged in text, we have responded by adding the following: *It is important to consider the impact of prior collaboration on results, although the participants may have had enhanced exposure to other technologies through prior work, they had not previously been involved in any VR studies.*

Very good to have a separate researcher conducting the developer interview regarding bias. However, in the focus groups, the duration of the interviews is mentioned, and it may be beneficial to include the interview duration in this section as well for consistency and additional context.

Thank you for this comment, we agree using the separate researcher will have helped against bias, and note that we missed the interview duration, this has now been added: *The interview was one hour in duration.*

Data collection When was the finding presented to the technology developer, prior to or during the interview?

Thank you for spotting this missing information – now amended: *For the one follow-up interview (approximately 10 months after the data collection and feedback),*

Data analysis P5 S45-53: This is a general description on the analysis. Did the nine researchers perform all steps?

All 9 researchers conducted initial coding, then HB, LC and RB led on iterations of reviewing and defining themes, then all researchers joined together again to explore perceptions on interpretation. I have clarified this in the text now.

How is the unit of meaning defined? HB, LC and RB led on iterations of reviewing and defining themes. However, it is unclear how the reliability of the codes (57 initial codes) was ensured in this study with multiple researchers analyzing the data. Were the codes mutually exclusive? Were all

themes covered in all workshops? Since this research follows an inductive approach, did the interview guide undergo any modifications during the data collection process?

Thank you for these thoughts, the unit of meaning was generally sentences, to ensure context and meaning wasn't lost by using smaller units. The reliability of the codes was ensured through meetings between all 9 researchers to review and discuss coding, ensuring team agreement on interpretation. All themes were covered across the entire dataset, with codes from each data set being used within the themes, although codes were worded differently between researchers, the meetings allowed us to refine and bring together those with shared meaning. The interview was semi-structured and therefore there was some natural deviation from the set interview schedule in terms of natural conversation and follow-up questions on interesting comments. This detail has been added to the manuscript.

Participants This might fit better earlier in the methods section.

We understand some people do include participants information earlier in the Methods section, however we disagree in this instance. The Participants section is within the Methods section, just ahead of the Results section to give context to the following findings.

Results (or findings?) Section 1: Workshop results - In Table 1, only two of the eight sites are described as Workshop.

Thank you, now amended.

P6 S50-57. The introduction of Section 1 and 2 might fit better under the Results heading than the Section 1 heading.

We understand your comment but feel the sections within the Results section are important to differentiate between the two types of data we're reporting on.

Table 2 – This can be part of the methods section and referred to in the Data analysis.

Table 2 directly reports the analysis of results and is therefore more appropriate remaining in the results section rather than the methods section.

Section 2: Results of the follow-up interview with the technology developer There is a significant amount of space dedicated to the results from Part 2, which is not actually mentioned in the study's title or included in the objectives stated in the abstract.

Thank you for this point, we have added the detail to the title.

Discussion The results are divided into two sections, and it would be beneficial to structure the discussion accordingly. Further exploration is warranted on the results pertaining to physical activity and exercises for older adults, as well as how staff members can effectively implement these interventions. Additionally, it would be valuable to delve into the level of technical competence required for staff to successfully conduct such sessions with older adults.

Thank you, we have added the following to the discussion: *Further exploration is also required to quantitatively assess the impact of the physical activity (or motivation to exercise) over longitudinal periods on outcomes for older adults. Additional future research should also seek to document the technical competence required for care staff to implement and facilitate such technologies without researchers present.*

Strengths and limitations are thoroughly addressed in the study.

Thanks very much for this comment and all of the useful feedback you have provided.

Reviewer: 2

Comments to the Author:

This manuscript explores how to improve the physical activity of the elderly in nursing homes through the views and feedback of 44 healthcare professionals on ROVR. I think it is very interesting and practical. However, I have some suggestions for improving the quality of the manuscript that I hope the authors can seriously consider.

Thank you very much for this comment, we are pleased you consider the manuscript interesting.

Q1. In my experience, even healthy older people may wear a headset VR display for a bit longer (maybe more than 20-30 minutes?). Symptoms like dizziness or nausea are very noticeable. The number of elderly people suffering from chronic diseases in nursing homes is very large. In practice, how to achieve exercise intensity while keeping patients comfortable is a worthwhile challenge.

Thank you, we have added the following to the discussion: *Future research should consider how to achieve the intended exercise intensity while ensuring resident comfort over suitable durations of VR use.*

Q2. I would like to know whether the author took into account the different feedback that may be caused by different levels of health management work environment and different subprofessional directions when conducting the survey of health management staff. In other words, can the 44 employees surveyed comprehensively cover the real thoughts of people in the same industry?

Thank you for this comment, we've added the following into the manuscript in response: *The participants included provided a broad range of appropriate stakeholders for English care homes, including across all levels of the operational care home staff and additionally various health professionals with expertise relevant for movement and activity (e.g. occupational therapist and physiotherapist).*

Q3. The results are not clear due to the long content of the results. It is suggested that the authors simplify the relevant content to improve the readability of this manuscript.

We understand this concern, however, the results sections of qualitative papers are often longer owing to the nature of the results as chunks of text. We would prefer to keep the results section in full as this gives the most useful information to the reader and also allows transparency on our interpretation by demonstrating direct quotes.

VERSION 2 – REVIEW

REVIEWER	Gjesdal, Beate Eltarvåg Western Norway University of Applied Sciences
REVIEW RETURNED	30-Aug-2023
GENERAL COMMENTS	Thank you for the opportunity to read this improved revised version. My recommendation is to tighten the use of terminology and refine the methodology.

The entire manuscript could benefit from tightening up the use of terminology. Some examples follow:

- For example, the heading "Intervention" is only used as a heading in the abstract, but it doesn't appear anywhere else in the manuscript.
- "Feedback" is used in several settings. Is the feedback from individuals working with care residents, and the feedback they provide to the system developer, referred to as feedback given to the system developer? Clarification of the type of feedback throughout the manuscript (e.g., p6 s42) would be helpful.
- "Results" and "findings" are used interchangeably.
- In Table 2, is the evidence provided to support transparency, or to show transparency?

Data Collection:

- What about the feedback that was raised during the interviews? Were they analyzed alongside the other materials?

Data Analysis:

- The initial 8.5 lines within the data analysis section could be found in any qualitative article utilizing thematic analysis. This is an opportunity to demonstrate transparency in the process. Just as the technical equipment is meticulously described, I am missing information about the specific steps undertaken in the analysis.
- The analysis was conducted inductively, despite the researchers having pre-determined aims. Further elaboration on the inductive nature would be beneficial.
- It's mentioned that "All nine researchers conducted initial coding" – does this imply that each researcher coded the entire dataset, or was it divided in some way?
- This could also be an appropriate juncture to discuss data saturation, considering the potential challenges posed by a substantial dataset and multiple coders.
- You mention that the research was strengthened by a multidisciplinary team – I concur, but how precisely has this bolstered your study?
- All researchers involved either possess or are working towards a PhD – does this imply a strong familiarity with qualitative methods?

Results could be presented more cohesively. For instance, the authors employ the phrase "Participants noted" three times within the "Anticipated Benefits" section.

Discussion:

	 • The discussion commences with an "aim" that does not align with the "Aims" outlined earlier in the article. Here are some comments on the wording in the manuscript:  • P2 s 31: The abstract states that it's an evaluation study, but the aim of the study suggests that it aims to explore staff perceptions. Clarification is needed. • P3 s7: Are you certain about using the term "significant" in qualitative research results? • P3 s30: The manuscript mentions that having a large dataset is a strength due to data saturation, but this isn't discussed elsewhere in the manuscript, despite being indicated in COREQ. • P3 s60: Is this where you intend to present the study's background? • P5 s15: The sentence ends with "design" (which is the heading). To explicitly refer to the equipment's design, it could be specified. • P6 s3: The procedure includes recruiting and workshops. • P6 s17: "All participants provided written..." This is already mentioned under ethics and doesn't need to be repeated. • P6 s37: Did the participants from the University workshop have experience with home care? • P7 s18: Interview schedule or interview guide? • P7 s53: There are repetitions of "Participants," with some redundancy from the "Procedure" section.
--	---

VERSION 2 – AUTHOR RESPONSE

Dear Reviewer,

Thank you for reading our improved manuscript. We have now responded to your further recommended changes as detailed below. We hope these changes have helped with the flow of the manuscript and appreciate the time taken for both rounds of reviews.

On behalf of all authors, thank you,

Hannah Bradwell

The entire manuscript could benefit from tightening up the use of terminology. Some examples follow:

For example, the heading "Intervention" is only used as a heading in the abstract, but it doesn't appear anywhere else in the manuscript.

- The heading 'intervention' was added to the abstract in line with BMJ Open rules on abstract structure. We previously titled this section 'method' but this is not an option in the BMJ Open abstract author guidelines. We are happy to go with whichever heading the editor would prefer.

"Feedback" is used in several settings. Is the feedback from individuals working with care residents, and the feedback they provide to the system developer, referred to as feedback given to the system developer? Clarification of the type of feedback throughout the manuscript (e.g., p6 s42) would be helpful.

- Thank you. We have reviewed our use of the word "feedback," yes the feedback on the product is from the participants (care staff and health professionals), which in turn has been provided as feedback to the developers to use for design improvements. In all instances it refers to this 'feedback' (i.e. what the participants thought of the device). We feel consistent use of the word feedback makes sense, as the feedback remains the user-centred perceptions throughout, whether reported on from participants or how this is provided to the business. To help clarify, we have added to the manuscript in instances where we refer to feedback provided to the business, to confirm the feedback referred to is from the participants and the workshops.

"Results" and "findings" are used interchangeably.

- We have replaced instances of 'findings' with 'results' for consistency.

In Table 2, is the evidence provided to support transparency, or to show transparency?

- The evidence is provided to support the codes, ie. We seek to support our interpretation of the data into the codes/themes included by demonstrating the raw data in relation to them.

Data Collection: What about the feedback that was raised during the interviews? Were they analyzed alongside the other materials?

- We discuss the analysis of the interviews in the data analysis section: "The process for the analysis of the follow-up interview with the technology developer was the same"

Data Analysis:

The initial 8.5 lines within the data analysis section could be found in any qualitative article utilizing thematic analysis. This is an opportunity to demonstrate transparency in the process. Just as the technical equipment is meticulously described, I am missing information about the specific steps undertaken in the analysis.

- Thank you, we have edited these lines to be more specific to our process and hope this reads better as our process.

The analysis was conducted inductively, despite the researchers having pre-determined aims. Further elaboration on the inductive nature would be beneficial.

- Thank you, we have added a sentence to expand on this.

It's mentioned that "All nine researchers conducted initial coding" – does this imply that each researcher coded the entire dataset, or was it divided in some way?

- The transcripts were divided between researchers, this has been clarified in text, thank you for noting this.

This could also be an appropriate juncture to discuss data saturation, considering the potential challenges posed by a substantial dataset and multiple coders.

- Thank you, we have not added discussion on data saturation, as each researcher completed their allocated transcripts, we did not seek to identify saturation at the time of coding.

You mention that the research was strengthened by a multidisciplinary team – I concur, but how precisely has this bolstered your study?

- Thank you, we have added the sentence: "The range of experience provided ensured broad perspectives were considered in our interpretations."

All researchers involved either possess or are working towards a PhD – does this imply a strong familiarity with qualitative methods?

- Yes, thank you, we have added this to the text.

Results could be presented more cohesively. For instance, the authors employ the phrase "Participants noted" three times within the "Anticipated Benefits" section.

- We are happy with our use of 'participants noted' to help explain the participants reported perceptions.

Discussion:

The discussion commences with an "aim" that does not align with the "Aims" outlined earlier in the article. Here are some comments on the wording in the manuscript:

- Thank you, we have clarified in the discussion.

P2 s 31: The abstract states that it's an evaluation study, but the aim of the study suggests that it aims to explore staff perceptions. Clarification is needed.

- The abstract states the study is a qualitative evaluation, which is met via exploring the staff perceptions qualitatively. Exploring staff perceptions is the qualitative evaluation of their perceptions towards the device. We have met the word limit of the abstract and cannot add further words to clarify this any more.

P3 s7: Are you certain about using the term "significant" in qualitative research results?

- We are comfortable with using significant, it is not related to quantitative results which would be improper but used in the general use of the word significant.

P3 s30: The manuscript mentions that having a large dataset is a strength due to data saturation, but this isn't discussed elsewhere in the manuscript, despite being indicated in COREQ.

- The large sample is mentioned three times in total, and hopefully readers can gain an understanding of the size of the data set upon which our interpretation is based.

P3 s60: Is this where you intend to present the study's background?

- Apologies, we're unsure on the reviewer's question.

P5 s15: The sentence ends with "design" (which is the heading). To explicitly refer to the equipment's design, it could be specified.

- We are comfortable with the use of the word 'design' to refer to the device design.

P6 s3: The procedure includes recruiting and workshops.

- Thank you, we have split the procedure into recruitment and workshops.

P6 s17: "All participants provided written..." This is already mentioned under ethics and doesn't need to be repeated.

- *Thank you, this has been removed.*

P6 s37: Did the participants from the University workshop have experience with home care?

- *Thank you, this has been clarified.*

P7 s18: Interview schedule or interview guide?

- *We are happy with reference to the interview schedule.*

P7 s53: There are repetitions of "Participants," with some redundancy from the "Procedure" section.

- *Thank you, we acknowledge this point but are happy with the current use of 'participants' for clarity in this section.*